# Wireless Monitoring of Biological Objects at Microwaves

Irina Vendik [1,*], Orest Vendik [1], Vladimir Pleskachev [2] , Irina Munina [1], Pavel Turalchuk [1] and Vitalii Kirillov [1]

[1] Department of Microelectronics and Radioengeneering, St. Petersburg Electrotechnical University "LETI", 197376 Saint Petersburg, Russia; ogvendik@rambler.ru (O.V.); ivmunina@etu.ru (I.M.); paturalchuk@etu.ru (P.T.); vvkirillov@etu.ru (V.K.)

[2] "SIMICON" Ltd., 195009 Saint Petersburg, Russia; vvpleskachev@etu.ru

\* Correspondence: ibvendik@rambler.ru

**Abstract:** Electromagnetic (EM) wave propagation inside and along the surface of the human body is the subject of active research in the field of biomedical applications of microwaves. This research area is the basis for wireless monitoring of biological object parameters and characteristics. Solutions to the following problems are crucial for achieving the stated goals in the area of wireless monitoring: EM wave propagation inside and on-body surface. The biological object monitoring is based on a consideration of the following problems: (i) dielectric properties of a biological issue; (ii) EM wave propagation in biological medium; (iii) propagation of EM waves across the boundary of two media (biological medium–air): wave reflection and refraction; (iv) EM wave propagation in a multilayer biological medium; (v) EM wave propagation along the plane or curved surface of biological objects.

**Keywords:** electromagnetic wave; microwave frequency range; biological medium; wave propagation; reflection; refraction; surface wave; creeping wave; multilayered biological medium; body area networks



## 1. Introduction

Systems of radio-frequency identification (RFID) are used for biomedical applications, such as remote diagnostics and wireless monitoring of human health. A great number of publications [1–4] confirms great interest directed to this field of research. The following problems are of high importance: (i) propagation of electromagnetic (EM) waves inside the biological medium, (ii) EM wave propagation along the interface between different media, (iii) on-body surface electromagnetic wave propagation, and (iv) RF system design for biological object monitoring.

Much attention has been paid in recent years to RFID systems intended for biomedical applications, such as remote diagnostics of disease, wireless monitoring of human health, providing safety of life, etc. The RFID system consists of radio tags and readers connected with antennas and an information processing system. The peculiarities of the development of RFID tags for these applications are their miniature size and the ability to read information from the RFID tag, taking into account the properties of biological tissues considered as dielectrics with high dielectric permittivity and significant losses of the EM signal.

Two types of RFID systems are in use: near-field and far-field systems. For the near field system, magnetic coupling is used for radio tags and readers, limiting the influence of the dielectric properties of the medium on the signal transmission. For another type of RFID systems, the radiation of electromagnetic waves in the far zone is used, providing a significant range of radio wave propagation when the radio tags are attached externally to clothing or directly to the body surface for use in emergency situations. In this case, the dielectric properties of the biological medium are of high importance.

Among health monitoring systems, wearable and implanted systems are widely used. In the former case, RFID tags are attached to clothes or are placed on the human body surface. In the latter case, RFID tags are implanted into the human body. Figure 1 shows communication between wearable and implanted devices and an external base station [5].

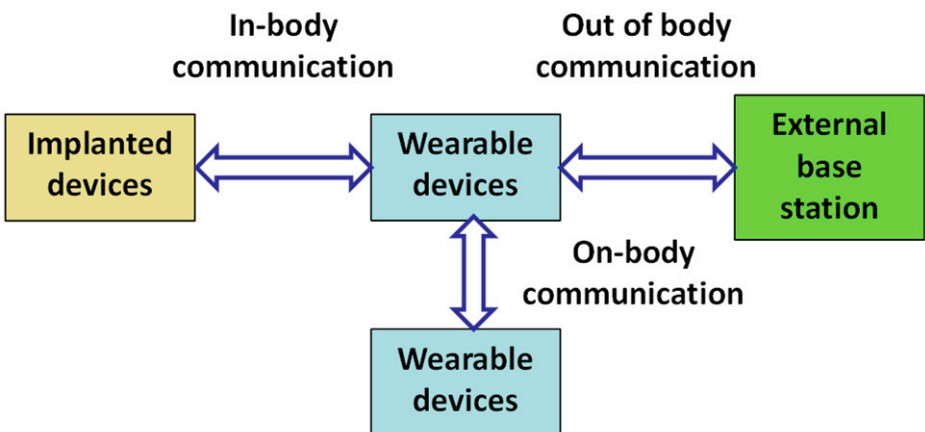

**Figure 1.** The system for collecting and transmitting information about the state of the human body. The system includes implanted and wearable devices and an external base station receiving collected information.

The development of wearable wireless systems is in wide demand due to a wide spectrum of applications such as personal communication, medicine, firefighting, radio frequency identification etc. Wearable electronics are very attractive due to the light weight, low fabrication cost, easy manufacturing, and availability of inexpensive materials. A new branch of wireless system technology is the Wireless Body Area Network (WBAN). Such systems, in combination with personal area networks, provide monitoring of the state of the biological systems (human body) in real-time. For the estimation and control of the data transfer by using electromagnetic waves over the body surface, it is necessary to describe properly the EM wave propagation. The interface between the air and the body surface supports three basic EM wave modes: surface wave, leaky wave, and creeping wave [6]. The first two modes are well studied [7,8], while the latter [9] exhibits a specific property on curved parts of the body and "creeping" into shadow regions [9,10].

Solving problems of the biological object monitoring, specific processes deserve thorough consideration:

- Characteristics of EM waves propagating in a biological medium: phase velocity and attenuation of the wave.
- Penetration of the EM waves through the interface between the biological medium and air (free space); the process is characterized by wave reflection and refraction.
- Propagation of the EM wave in the multilayer biological object.
- On-body surface electromagnetic wave propagation.

## 2. Electromagnetic Waves in Biological Medium: Parameters and Characteristics

Bio-sensors placed inside the human body or on its surface are used for transferring information about the state of the body by means of EM waves and provide body–environment coupling, presenting information about processes in the body being observed in real-time. Temperature and blood pressure sensors, control of heartbeat, muscle tension, broken bone recovery maintenance, etc., are examples of implanted devices. The implanted device introduces collected data into a system that modulates the radiated EM wave and thereby provides data transfer to the environment. A wearable device may gather data for the state of the human body and/or transfer them to the environment or serve as an antenna coupled with an external base station. The coupling is provided by EM waves propagating between the sensor inside or over the body and the external antenna. Propagation in the homogeneous tissue as well as in the layered medium is under consideration.

### 2.1. Propagation of the EM Wave in a Biological Medium

The biological medium behaves as a dielectric material described by the dielectric permittivity and conductivity. The low-frequency permittivity of the human tissue is formed by the presence of macromolecules, cells, and other components coupled by membranes. At low frequencies ($f < 1$ MHz), the membranes are of high capacitance. At frequencies near 100 MHz, the capacitance of the membranes is determined by the rotation and vibration properties of polar molecules. These properties are responsible for a high dielectric permittivity. A high dielectric loss is provided by a high conductivity of biological tissues. As a rule, the permittivity of the biological medium decreases and its conductivity grows with frequency increasing [2].

The most common models used to describe the dielectric properties of biological tissues are: Debye, Cole–Cole, and Cole–Davidson models [11].

The frequency dependence of the permittivity can be analytically described by the following formula [12]:

$$\varepsilon(\omega) = \varepsilon_\infty + \frac{\varepsilon_s - \varepsilon_\infty}{\left[1 + (i\omega\tau)^{1-\alpha}\right]^\beta} + \frac{\sigma_s}{i\omega\varepsilon_0} = \varepsilon'(\omega) - i\varepsilon''(\omega). \tag{1}$$

Here, $\omega$ is the angular frequency of the electric field; $\varepsilon_\infty$ is the dielectric permittivity at the frequency $\omega \to \infty$, caused by electron polarizability; $\varepsilon_s$ is the low-frequency permittivity; $\sigma_s$ is the low-frequency (static) conductivity, determined by the motion of charged particles; $\varepsilon_0$ is the free space permittivity; $\tau$ is the characteristic relaxation time of the medium. This parameter is determined as the time required for molecules or dipoles to return to their initial state, which was disturbed by the applied electric field. For $\alpha = 0$ and $\beta = 1$, the Equation (1) corresponds to the Debye model. For $0 < \alpha < 1$ and $\beta = 1$, the Equation (1) is consistent with the Cole–Cole model taking into account the relaxation time dispersion. For $\alpha = 0$ and $0 < \beta < 1$, Equation (1) corresponds to the Cole–Davidson model, which takes into account the asymmetric distribution of the relaxation time. Biological tissue is usually described by the Debye model and occasionally by the Cole–Cole model.

Fractional terms in (1) indicate the relaxation (non-resonance) type of the frequency-dependent permittivity, which consists of a real and an imaginary part. Instead of the imaginary part of the permittivity, the electrical conductivity of the medium $\sigma = \varepsilon_0\varepsilon''\omega + \sigma_0$ is used ($\sigma_0$ is the frequency-independent conductivity). The frequency dependence of the real and imaginary parts of the relative dielectric permittivity and conductivity calculated for an averaged biological tissue by formula (1) is given in Figure 2 [13].

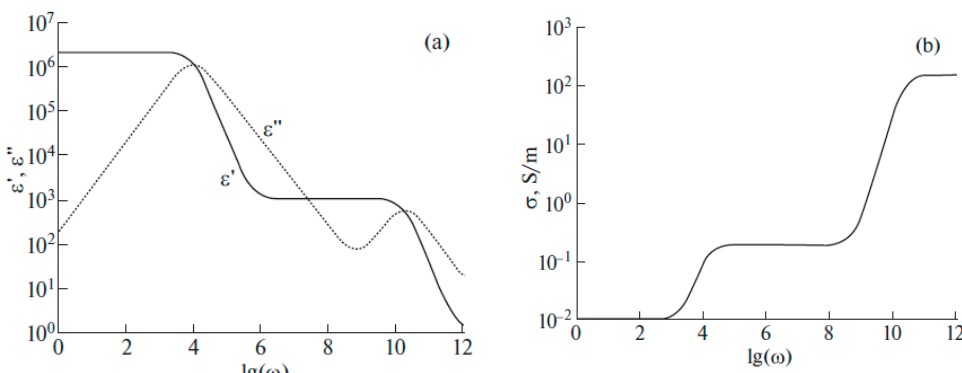

**Figure 2.** Frequency dependence of the real and imaginary parts of the relative dielectric permittivity (**a**) and conductivity (**b**) of biological tissue with averaged parameters calculated by the Debye formula.

The dielectric properties of a set of biological tissues were measured and analyzed in a wide frequency range (10–20 GHz) in [14,15]. The measurements were carried out in vitro on different tissue samples extracted from a living organism under fixed external

conditions. Further studies showed that there is a significant scattering in the results of measurements of the parameters of biological tissues. The spread is due to the use of various measuring techniques, as well as the sensitivity of tissues to changes in temperature, humidity, etc. In [16,17], parameter scattering was analyzed. Parameters of various tissues in the frequency range 2–10 GHz are presented in Table 1.

**Table 1.** Electric properties of different biological tissues.

| Tissue | Dielectric Permittivity | Conductivity, S/m |
|---|---|---|
| Skin | 28–45 | 1.00 |
| Muscle | 30–70 | 0.95–1.55 |
| Tendon | 46 | 1.10 |
| Fat | 2–6 | 0.05 |
| Cortical bone | 12 | 0.20 |
| Trabecular bone | 12–27 | 0.44–0.55 |

The propagation of EM waves is described by the Maxwell equations. The solution for the electric component $E$ of a plane EM wave propagating in the $z$ direction is

$$E(z) = E_m e^{-\alpha z} e^{-i\beta z} e^{i\omega t} \tag{2}$$

where the factors $e^{-\alpha z}$ and $e^{-i\beta z}$ describe, respectively, the attenuation and the phase advance of the wave, and $e^{i\omega t}$ describes the dependence of the phase of wave on time.

In general, the propagation of an EM wave is characterized by the complex wave number:

$$k(\omega) = \beta(\omega) - i\alpha(\omega) \tag{3}$$

where $\beta$ is the propagation constant, and $\alpha$ is the damping ratio. In the case of a small value of loss tangent ($\sigma \ll \omega\varepsilon''$), the parameters $\beta$ and $\alpha$ may be written as [12,13]:

$$\beta(\omega) = \omega\sqrt{\varepsilon_0 \varepsilon'(\omega)\mu_0} \tag{4}$$

$$\alpha(\omega) = \frac{\sigma(\omega)}{2}\sqrt{\frac{\mu_0}{\varepsilon_0 \varepsilon'(\omega)}} \tag{5}$$

The permeability of the biological tissue is equal to the permeability of the free space $\mu_0$. The phase velocity of the wave is determined by the propagation constant $\beta$:

$$V_{ph}(\omega) = \omega/\beta(\omega) \tag{6}$$

*2.2. Propagation of the EM Wave through the Biological Medium–Air Interface: Reflection and Refraction*

A small-size antenna implanted into the biological tissue is considered a source of EM wave. Unlike the simple case of wave propagation in a homogeneous isotropic medium, the EM wave propagation through the biological tissue–free space interface is accompanied by the existence of some specific problems [18–20].

A segment of the wave that is much smaller than the distance to the source position can be roughly considered as a plane wave (Figure 3a). The propagation of spherical EM waves with the angle of incidence $\theta$, considered far away from the source, can be described as the first approximation in terms of plane waves.

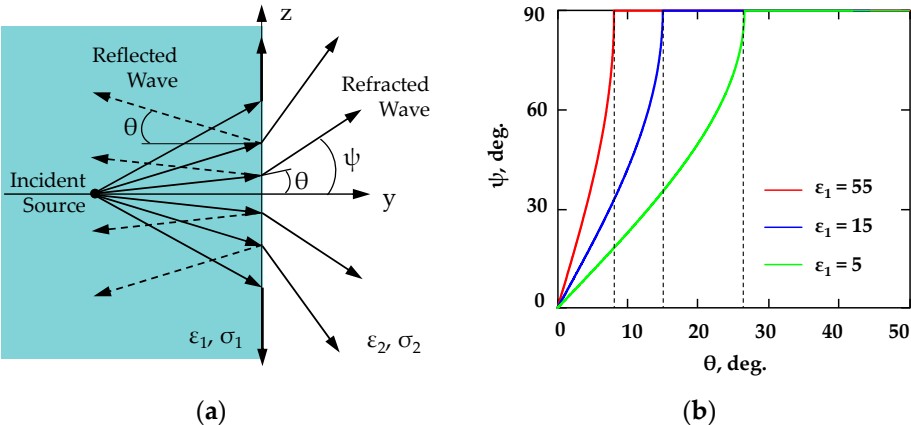

**Figure 3.** The spherical wave propagation across the interface between two dielectrics (**a**) and the dependence of the angle of refraction on the angle of incidence (**b**) for dielectrics with different values of permittivity ($\varepsilon_1$ = 55, 15, 5) and $\varepsilon_2$ = 1 [13,18].

The diffraction of a spherical wave on a flat interface between two dielectric media (Figure 3a) is characterized by the refraction angle $\psi$ of a plane EM wave related to the angle of incidence $\theta$ as [13]

$$\cos(\psi) = \sqrt{1 - \frac{\varepsilon_1}{\varepsilon_2} \sin^2(\theta)} \tag{7}$$

As an example, consider the EM wave propagation through the interface formed by a dielectric material with $\varepsilon_1$ = 55 (biological medium (muscle) at $f$ = 2.45 GHz) and air ($\varepsilon_2$ = 1). The deviation of the incident wave from the normal to the interface by more than 7.75° causes the deviation of the refracted wave from the normal by 90°. That means that the refracted wave does not propagate in the medium with $\varepsilon_2$ = 1 and goes along the interface surface (Figure 3b). The angle of incidence $\theta_0$ at which the refracted wave does not propagate in the environment is known as the angle of total internal reflection. That means that only a small fraction of the power flux radiated by the source lying in the sector of ±7.7° enters the surrounding space, whereas most of the radiated energy is transformed into a surface wave at the interface. The higher the dielectric contrast at the interface, the smaller portion of the radiated energy is able to penetrate into the air.

### 2.3. Propagation of the EM Wave through the Interface of Biological Tissue and Free Space

In the case of using a device (sensor) implanted in the biological medium, the degradation assessment of the electromagnetic signal is of high importance. The high loss factor leads to the attenuation of a signal in the medium. Additionally, high contrast of the dielectric permittivity of the biological medium and the free space results in a strong wave reflection from the interface. It is important to study the EM wave propagation in a biological medium, including the refraction of waves at the biological tissue–free space interface [21–23].

Generally, for electromagnetic waves propagating toward the interface between the different dielectric media, the effects of reflection and refraction occur at the interface. Taking into account the dielectric loss in the biological tissue, the attenuation of the wave caused by the refraction of the wave propagating through the boundary between two dielectrics is defined as [13,18,19,23].

$$A_{refr} = -10 \log\left(\frac{P_2}{P_\Sigma} e^{-\alpha h}\right) \tag{8}$$

In Equation (8), the energy of reflected wave is not taken into account.

Here, $\alpha$ is the attenuation coefficient determined by Equation (5); $h$ is the distance from the source to the boundary surface; $P_\Sigma$ is the power radiated by the implanted antenna into the biological medium and for a dipole with $l = \lambda/2$ is defined as

$$P_\Sigma = \eta_1 \frac{|I_0|^2}{4\pi} \int\limits_0^\pi \frac{\cos^2\left(\frac{\pi}{2}\cos\theta\right)}{\sin\theta} d\theta \qquad (9)$$

here $\eta_1 = \eta/\sqrt{\varepsilon_r}$, $\eta = 120\pi\,\Omega$ is the wave impedance of the free space, and $I_0$ is the current amplitude.

$P_2$ is the power radiated by the antenna into the free space (air), which is limited by the angle of total internal reflection $\theta_0$:

$$P_2 = \eta \frac{|I_0|^2}{4\pi} \int\limits_0^{\theta_0} \frac{\cos^2\left(\frac{\pi}{2}\cos\theta\right)}{\sin\theta} d\theta \qquad (10)$$

Power transmission from the miniature antenna inside the biological tissue and the external antenna placed outside the tissue were found by the simulation using the full-wave electromagnetic simulator SEMCAD X by SPEAG [24]. The transmission coefficient $S_{21}$ was estimated for a typical structure containing a dipole implanted in the tissue with the dielectric properties typical for the biological tissue. The scheme of the experimental setup used for measuring the response of the implanted antenna is shown in Figure 4.

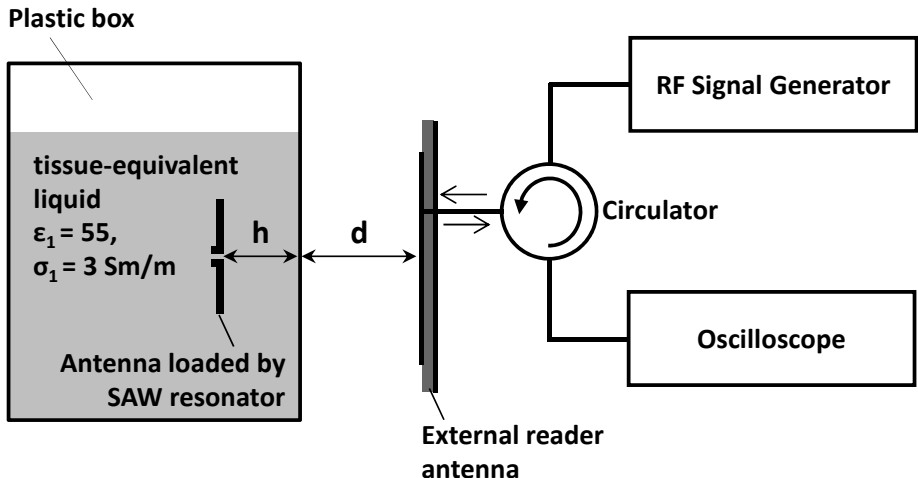

**Figure 4.** The scheme of the experimental setup for measurement of the transmission coefficient of the wave propagating through the interface.

In order to verify the simulation results, the implanted antenna has been fabricated and embedded inside the liquid with dielectric properties similar to the characteristics of the biological tissue ($\varepsilon_1 = 55$ and $\sigma_1 = 1.5\,\text{S/m}$).

The implanted antenna was designed as a folded dipole (Figure 5a). The dipole was loaded by a sensor designed as the surface acoustic wave (SAW) resonator. The SAW resonator implemented with the 50 Ohms output impedance is connected directly with the antenna terminals by gold wires. The tag based on the SAW element is the temperature-dependent sensor used for wireless measurements. The tag was embedded inside the liquid equivalent of the biological tissue.

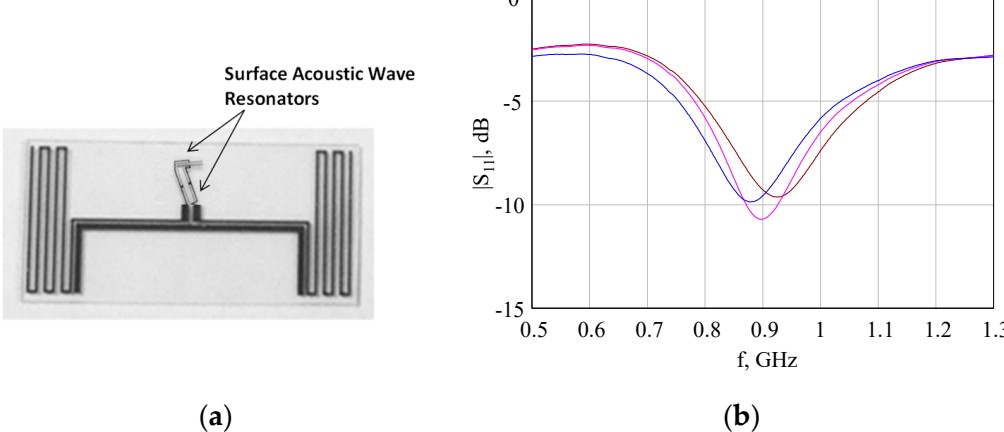

(**a**)                                         (**b**)

**Figure 5.** (**a**) The design of the implanted dipole loaded by SAW resonator; (**b**)return loss measured for three test samples of the implanted dipole in the tissue-equivalent liquid.

The dipole with an area of $20 \times 10$ cm was fabricated on the quartz substrate using the photolithography process. The surface of the substrate with 0.5 mm of thickness was covered by a plating layer of copper (4 μm) using chromium sub-layer. The measured return loss of the fabricated dipole embedded into the plastic box filled with the solution of sodium chloride and Ethanol with dielectric properties similar to the properties of a biological medium is shown in Figure 5b.

The result of the simulation of the transmission coefficient $S_{21}$ as a function of the distance $h$ between the dipole and the tissue surface at the frequency $f = 915$ MHz (ISM band) is presented in Figure 6a. The simulation results are presented together with the results of calculation using Equations (8)–(10) additionally, considering the path loss and medium mismatching. The electric field distribution in the structure under consideration is shown in Figure 6b.

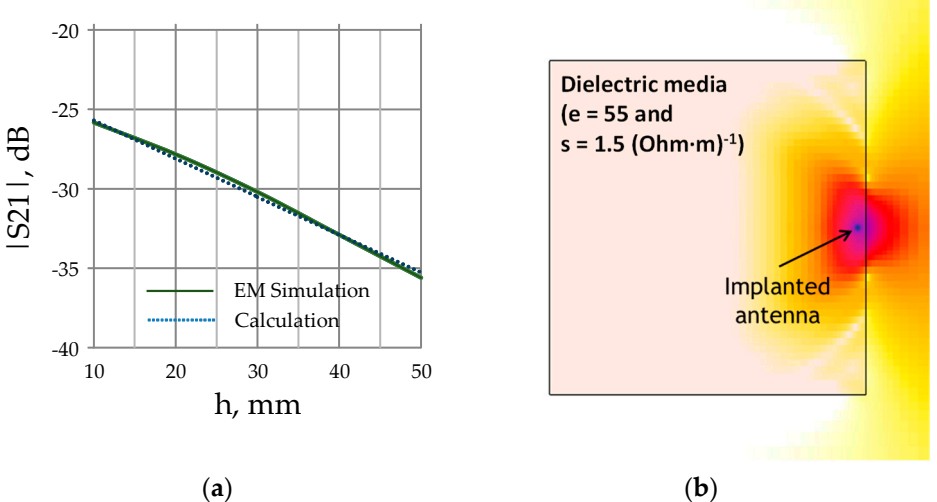

(**a**)                                         (**b**)

**Figure 6.** (**a**) Transmission coefficient module $|S_{21}|$ versus distance between the implanted dipole and the dielectric medium surface $h$. (**b**)The electric field distribution radiated by an antenna inside the dielectric medium.

The RF analog signal generator Agilent N5181A with a signal power level of 23 dBm was used for irradiation of the implanted antenna. For the registration of the backscattered response of the antenna, the oscilloscope Agilent Technologies Infiniium DSO80304B was used. The resonant frequency 907 MH of the SAW sensor response has been observed. The measured relative dielectric permittivity and the conductivity of the tissue-equivalent

liquid are $\varepsilon_r = 55$ and $\sigma = 3$ S/m, respectively, which is close to the parameters predicted by the simulation.

The backscattered response power level of the folded dipole antenna with the SAW sensor as a function of the distance between the reader antenna and the tissue-equivalent liquid box surface *d* and of the distance between the implanted sensor and the tissue surface *h* are shown in Figure 7.

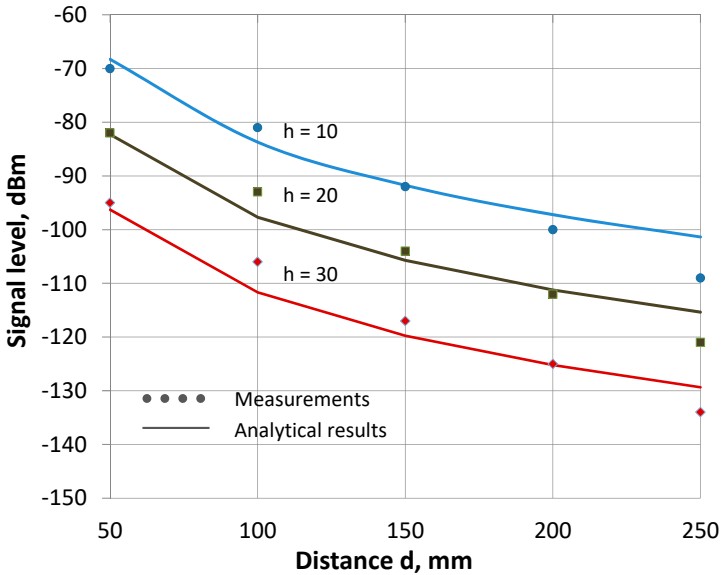

**Figure 7.** Measured signal power level versus distance *d* between the reader antenna and the box with the tissue-equivalent liquid for different values of distance *h* between the implanted dipole and the tissue surface. The results of calculation by (8)–(10), taking into account the path loss and medium mismatching, are shown by solid lines, and the dots present the measured results.

Evidently, the EM wave is remarkably reflected and refracted on the boundary between the biological tissue and the air (Figure 6b). The power level received by the reader antenna is low in the case of using implanted sensors. From this point of view, the on-body location of RFID sensors is preferable.

### 2.4. Attenuation of EM Wave Propagating through the Biological Tissue–Air Interface in Case of Using Matching Layer

EM wave propagating through the biological tissue–air interface is strongly suppressed. There are many reasons leading to this effect. The high contrast of the dielectric permittivity of the biological medium and the free space (air) results in a strong wave reflection from the interface and the wave refraction at the biological tissue–free space interface (Figure 3a). The high loss factor provided by the conductivity of the biological medium leads to attenuation of a signal in the medium. It is useful to analyze the possibility of increasing the EM wave transmission coefficient between the implanted tag antenna and the reader antenna.

Using a matching layer placed between two dielectric layers decreases the reflection coefficient of the EM wave from the interface. The dielectric permittivity of the matching layer can be calculated by the well-known formula [7]:

$$\varepsilon_{tr} = \sqrt{\varepsilon_{bio}\varepsilon_{air}} \tag{11}$$

where $\varepsilon_{bio}$ and $\varepsilon_{air}$ are the dielectric permittivity of the bio-tissue and free space, correspondingly, and $\varepsilon_{tr}$ is the dielectric permittivity of the matching layer. The thickness of the layer is equal to a quarter of the wavelength guided in the matching layer at the operational frequency.

For example, let us estimate the effectiveness of this approach for muscle tissue. In line with the experimental data presented in [14], the dielectric permittivity of muscle is $\varepsilon_r = 55$, and the electric conductivity is $\sigma = 1\,\text{S/m}$ at frequency $f = 0.915\,\text{GHz}$. The structure under investigation is shown in Figure 8a. The system with the following parameters was analyzed: the distance between the surface of the tissue and implanted antenna $h = 10\,\text{mm}$ and the distance between the surface of the tissue and the reader antenna $d = 50\,\text{mm}$. The simulation was performed by FDTD method using full-wave electromagnetic simulator SEMCAD X by SPEAG [24].

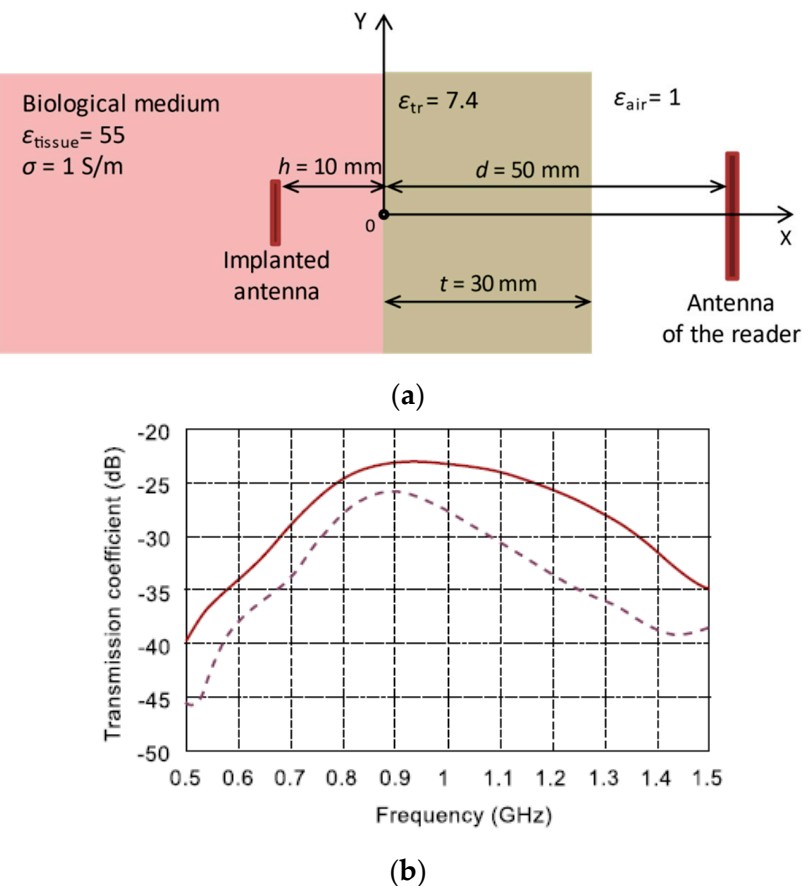

**(a)**

**(b)**

**Figure 8.** (**a**) A scheme of the structure containing the implanted antenna with matching layer placed between the biological tissue and the reader antenna; (**b**) the transmission coefficient between the antennas: Transmission coefficient is shown by the solid line for the structure with matching layer and by the dashed line without it.

The results of the EM wave propagation modeling for the system with two antennas, including the matching layer, are shown in Figure 8b. At 0.915 GHz, the dielectric permittivity of the layer is $\varepsilon_{tr} = 7.34$ and the thickness $t = 30\,\text{mm}$. Comparison of the results of electromagnetic simulation for structure with and without matching layer shows that the presence of the matching layer improves $S_{21}$ up to 3 dB. The E-field distribution along and across $x$-axis is shown in Figure 9. Using the matching layer improves the transmission coefficient from $-26$ dB up to $-23$ dB at the operational frequency $f = 0.915\,\text{GHz}$. Using the matching layer looks promising in the case when the implanted antenna is placed deep inside the tissue and the EM wave strongly attenuates in the biological medium.

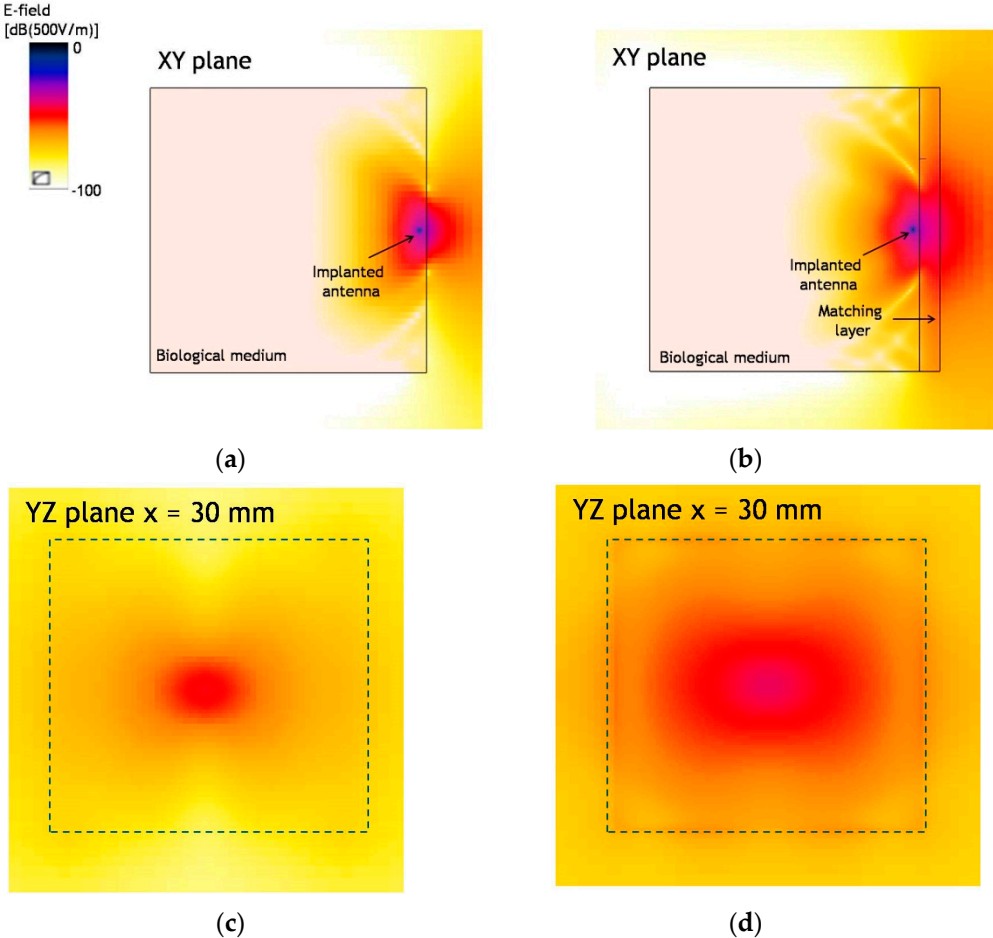

**Figure 9.** The E-field distribution: XY plane without (**a**) and with matching layer (**b**); YZ plane without (**c**) and with (**d**) matching layer.

### 2.5. EM Wave Propagation inside the Human Body

Propagation of the electromagnetic waves inside and around the outer surface of the human body is the subject of extensive research that bridges electrodynamics and medical science. This area of research lays the foundation for non-invasive biological object monitoring [13]. In contrast with the widespread X-ray diagnostic method, low-power microwave radiation is harmless for the human body if an adequately safe microwave signal power level is applied. Electromagnetic wave propagation through the various human tissues is the subject of this research which helps to detect early signs of disease and to diagnose the degree of tissue damage. The matter is complicated by the fact that the different parts of the human body contain many tissue layers with individual dielectric properties and geometry specific to a particular person. Hence the research goal is the in vivo estimation of the particular tissue state inside the multilayered human tissue structure.

Microwave imaging techniques and many other applications are utilized for extracting the tissue model parameters and registering harmful changes in the tissue condition due to the microwaves being able to penetrate inside the human tissues. The scattered or reflected signals from those tissues are measured and processed, giving useful insight into the underlying tissue structure. Microwave imaging system operates by observing contrast in the dielectric permittivity of various parts in the imaged objects and their surroundings. The microwave imaging technique is used for the examination of different tissues and organs of a patient: imaging of breast cancer [25,26], brain stroke diagnostics [27–29], microwave bone imaging [30–35], etc. In all cases mentioned, the multilayer dielectric

medium is investigated. It is important to analyze multiple reflections and refraction for many layer boundaries. Additionally, the EM wave attenuates in each layer depending on the conductivity, which is high enough for any tissue comprising the body. The main problem in the non-destructive analysis of the tissue is a low energy level of the EM wave reaching a target tissue and, as a result, a weak response signal makes it difficult to obtain a reliable observation.

Analyzing the EM wave propagating in the multilayered dielectric medium considered as the research object. As an example, a finger of a human hand is taken as a multilayered object composed of different tissues. The finger contains skin, fat, muscle, bone etc. Bone mineralization level may be used for the estimation of the state of the bone suffering from osteoporosis. When a patient is diseased with osteoporosis, the trabecular bone tissue is replaced by the marrow bone. The dielectric permittivity of the bone depends on the degree of the bone damage; therefore, the measurement results of the bone dielectric permittivity may be used as a diagnostic criterion estimating osteoporosis severity. The advantage of performing measurements on the finger is that the finger bones are covered by thin layers of different tissues, and therefore, the EM waves are given better access to the trabecular bone tissue.

A model of the finger containing two phalanges is shown in Figure 10. The finger model was developed by the authors to simulate EM wave propagating inside the finger tissues. The dimensions of the model components were picked as the average length, width, and height of the "average human" phalanges found inside the index finger. Two types of probes were used to perform simulations: an open-end coaxial probe [14,15,36–38] (Figure 11) and a microstrip antenna [32,33] printed on a dielectric substrate, which was applied on the finger (Figure 12a).

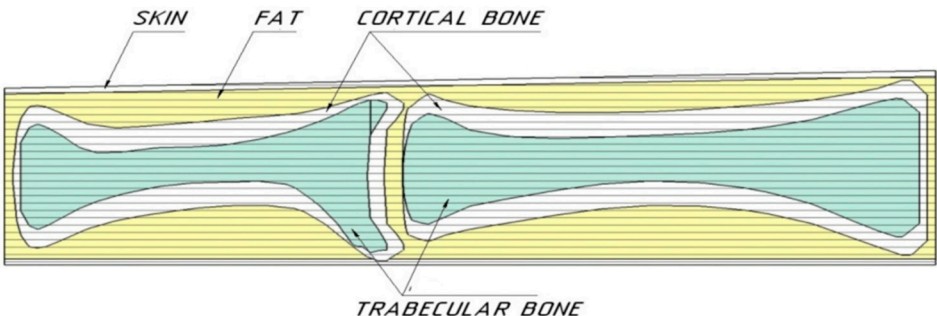

**Figure 10.** Cross-section of the finger model.

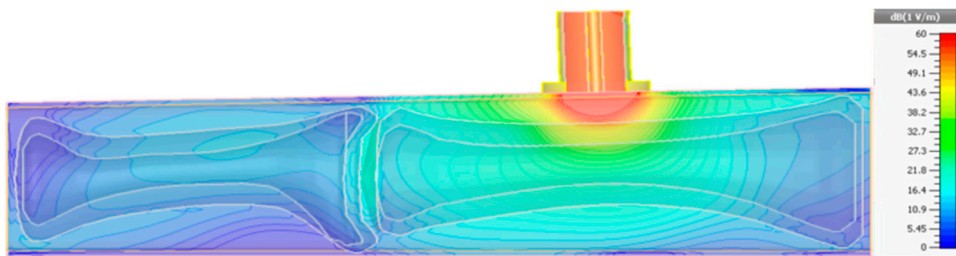

**Figure 11.** Electric field distribution within the finger volume excited by the open-end coaxial probe.

The electric field distribution in the case of using the open-end coaxial probe, obtained by the authors using full-wave simulation [39], is presented in Figure 11. Only a tiny fraction of the EM wave energy penetrates into the bone tissue due to intensive reflection and remarkable attenuation in the surrounding tissues. The field distribution may be also influenced by the wave propagating over the body surface [40,41].

The EM wave penetrates in the finger much more effectively if the finger is placed between a printed antenna and a conducting sheet simulated as the perfectly conducting

material (Figure 12a). Analysis of the frequency dependence of the reflection coefficient of the antenna revealed that two resonances were observed in the frequency range 0–5 GHz. The fact that the resonances are sharp enough facilitates their observation in the experiment. The wave excited inside the finger takes the form of several modes, as shown in Figure 12b,c. It is shown that the printed antenna is more efficient than the coaxial probe when it comes to EM field penetration inside the trabecular bone [40–44].

The printed antenna with dimensions of 5.5 × 0.35 cm was manufactured and tested on two persons: one of them is healthy, and the other has confirmed osteoporosis. Both EM field mode frequencies revealed in the experiment with the printed antenna applied on the finger are close enough to the modeling results as shown in Figure 12d despite the frequency shift around 600 MHz, which can be explained by the difference in tissue permittivity in modeling and real life. The mode at the frequency $f$ = 750 MHz is marginally sensitive to variations in the dielectric permittivity of the trabecular bone, whereas the mode with a higher frequency at 2225 MHz (2860 MHz in the experiment) shifts by about 140 MHz (340 MHz in the experiment) for the healthy and diseased persons. This fact could be used to predict osteoporosis. If the measurements are being performed regularly on a particular patient, the measured resonance frequency on the trabecular bone condition is obtained in dynamics. This systematic approach makes the measurements unsusceptible to the absolute value of the resonant frequency, which depends on finger geometry, individual tissue properties etc., as the measurement dynamics bring the personalized results for a particular patient.

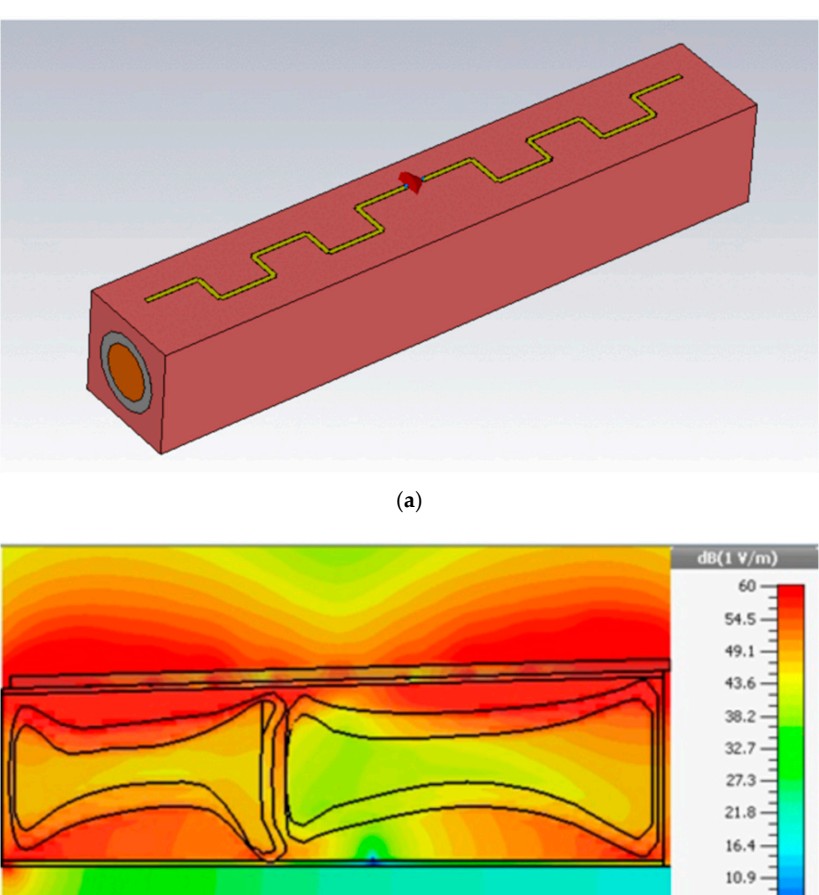

(a)

(b)

**Figure 12.** *Cont.*

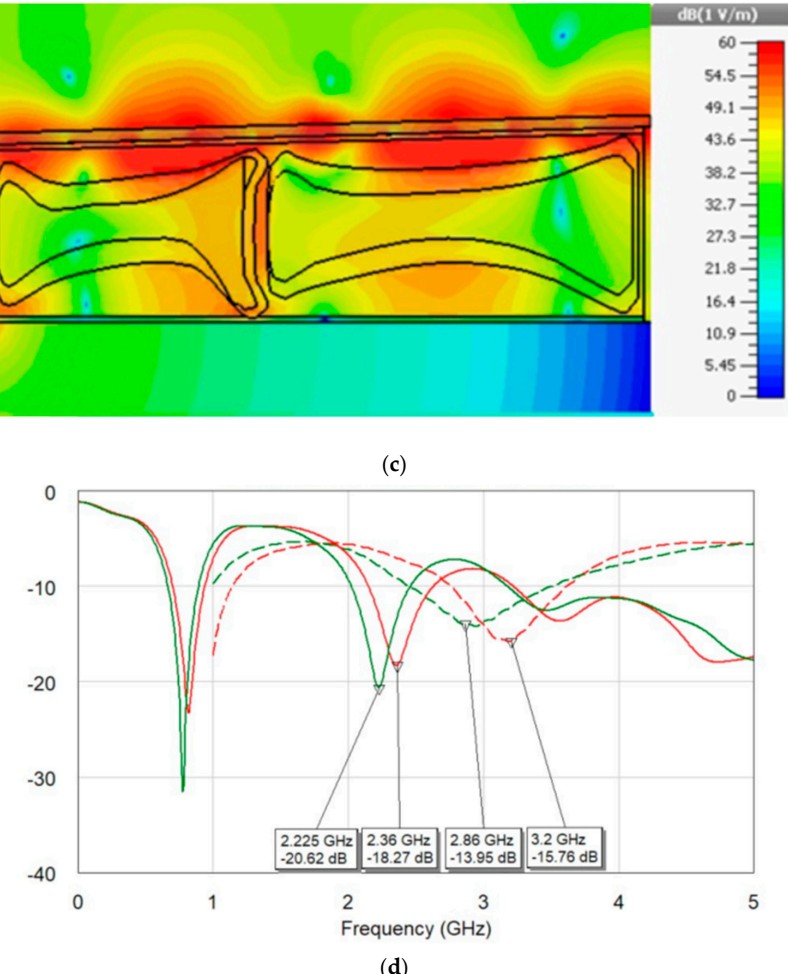

(c)

(d)

**Figure 12.** (**a**) Electric field distribution in the finger excited by the printed antenna: printed antenna placed on top of the finger, (**b**) electric field distribution inside the modeled tissues at 750 MHz and (**c**) at 2225 MHz; (**d**) return loss of the antenna: red curves—diseased bone, green curves—healthy bone, solid lines—modeling, and dashed lines—experiment [42,43].

## 3. Propagation of an EM Wave along the Surface of Biological Objects

The study of surface wave propagation mechanism is important in situations when the wireless data transfer between sensors placed on the human body surface is in need. In some situations, it is not feasible or impractical to connect by wire sensors such as biopotential or acoustic readers or execution devices such as injection pumps, with a computer running some software intended for interaction with the body. A good understanding of the wireless data channel properties is the key aspect of the wireless connection successful operation. Wireless links running on the surface of the human body have specific properties depending on the body dimensions, shape, and surface impedance, not to mention the influence of other factors such as individual variations in the surface impedance and presence or absence of clothes, etc. We do not address the specific properties that were mentioned trying to come up with general ideas on how the surface waves propagate. We start with the theory of the surface wave propagation briefly explained below and then contribute the results obtained to this research area.

The interface between two dielectric media with different values of the dielectric permittivity (body surface and air) allows for propagating electromagnetic waves along the interface radiated by an antenna. The interface between the air and the body surface supports three basic EM wave propagation modes [6]: surface wave (SW), leaky wave (LW) and creeping wave (CW). The SW and LW modes used at microwaves are well studied [7,8],

while the CW [9,40,41] demonstrate specific properties, propagating along rounded and curved parts and "creeping" into the shadow areas.

This particular property of the CW makes possible non-line-of-sight WBAN communications. Analytical description of the CW properties is studied in [9,10,25,26] and applied to wave propagation over a uniform cylinder.

The SW of the EM radiation is localized near the interface and propagates along the interface surface. The SW rapidly attenuates while propagating away from the interface in the direction normal to the interface.

The LW is radiated at some angle to the interface, which is defined by the permittivity ratio of two dielectric media. Finally, the CW propagates along the curved surface, falling into a shadow area (Figure 13).

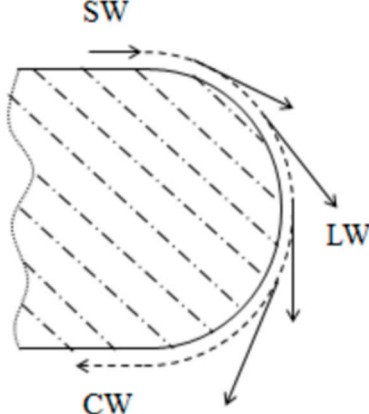

**Figure 13.** Types of waves on the surface of a complex shape.

Let us consider EM wave propagating over plane and curved surfaces of a dielectric object exhibiting dielectric properties of the human skin [13,42,43]. A simplified model of the object made of a uniform lossy dielectric or the perfectly conducting material was implemented to save full-wave electromagnetic simulation computation time [13,18,44]. The phantom made of lossy dielectric with $\varepsilon_r$ = 43.0 and $\sigma$ = 1.7 S/m is shown in Figure 14. It is formed by the rectangular box combined with a half-cylinder of radius $r$. The box represents the human chest and back with a flat surface, and the cylinder substitutes the forearm and shoulder. The overall phantom volume equals approx. 6000 cm$^3$ (Figure 14). The disk antenna is used to excite the wave propagating along the surface. The results of the simulation of the electric field distribution along the blue line are presented in Figure 14b. The surface wave field distribution is simulated over the flat surface while the CW distribution is kept on the cylindrical surface.

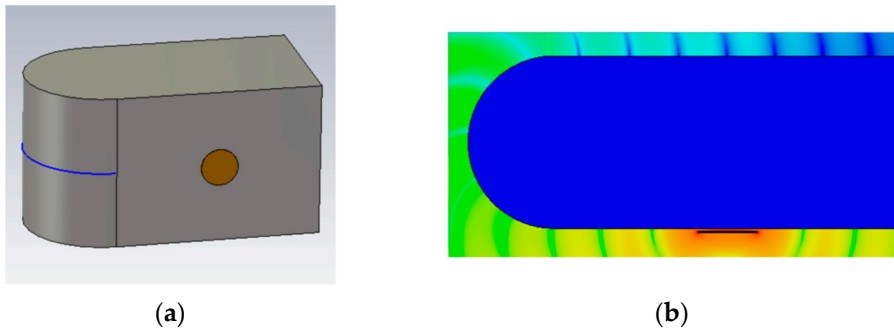

(**a**)                                        (**b**)

**Figure 14.** The model used for the surface and creeping wave simulation: (**a**) front sight; (**b**) the electric field distribution (top view).

Typical electric field distribution for the CW and SW along the phantom surface is shown in Figure 15.

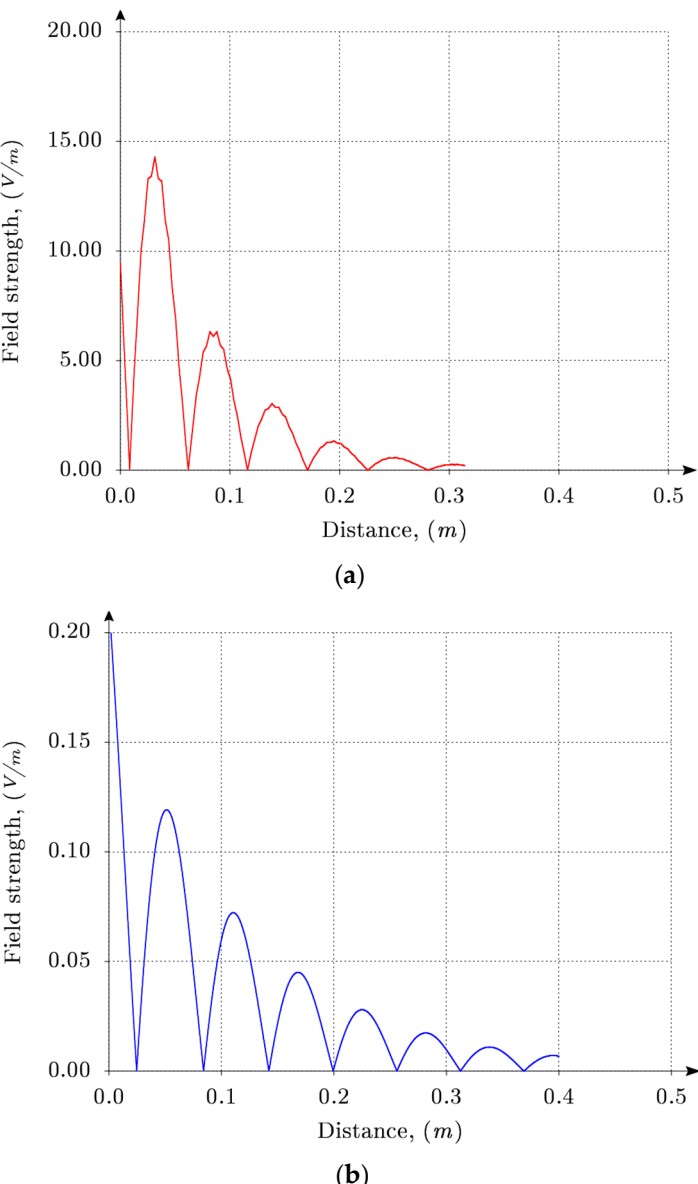

**Figure 15.** The distribution of the electric field component of the EM waves propagating along the surface at the frequency $f$ = 2.45 GHz: the creeping wave (**a**); the surface wave (**b**).

Using the simulated electric field distribution on the phantom surface (Figure 15), the wavelength could be calculated as $\lambda = 2l$, where $l$ is the distance between two adjacent field distribution minima. The phase velocity is defined as follows from [7]:

$$V_{ph} = f \cdot \lambda \tag{12}$$

where $f$ is the frequency of the EM wave propagating along the surface.

The frequency dependence of the phase velocity normalized to the speed of light in the free space for a set of different shoulder radius $r$ is presented in Figure 16.

The phantom thickness is $t = 2r$. From the presented data, one may conclude that the CW exhibits dispersion, whereas the SW is non-dispersive.

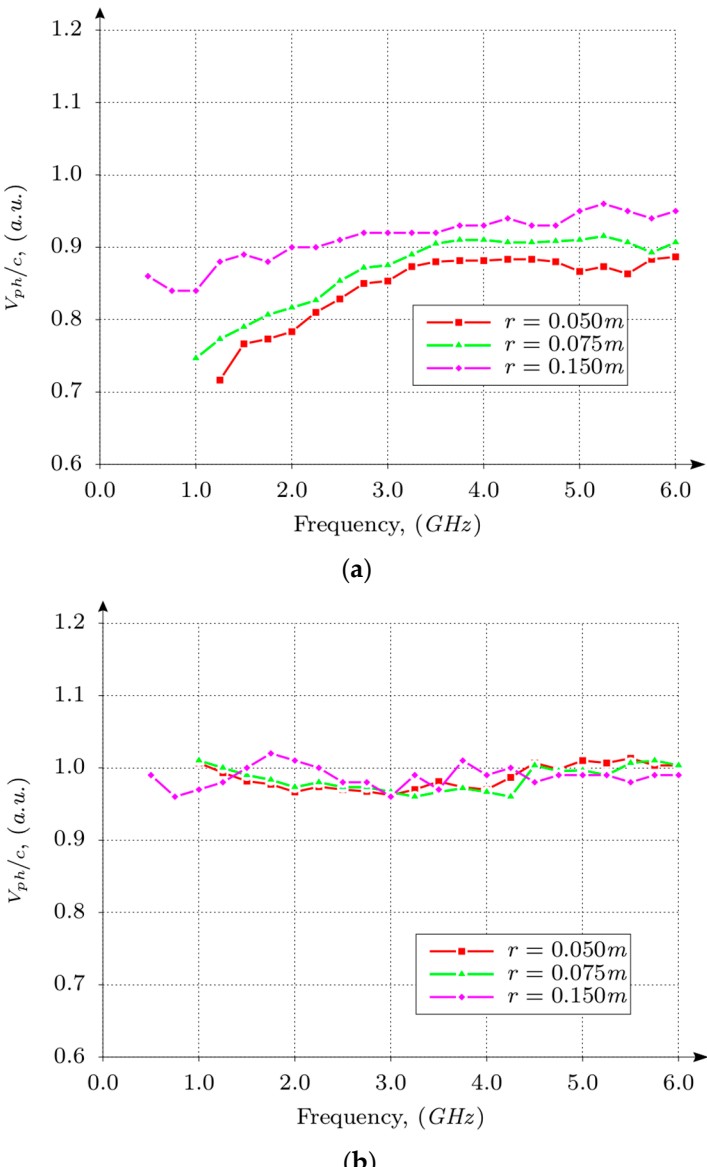

**Figure 16.** The normalized phase velocity of waves for different values of the phantom thickness: (**a**) creeping wave; (**b**) surface wave.

A simplified analytical model was proposed to calculate the CW phase velocity. For this purpose, a part of the cylinder with radius *r* is used to describe the surface of the phantom shoulder (Figure 17). The object is assumed to be made of the perfect conductor. This assumption is explained by a high value of the dielectric constant of the human tissue at the surface-air interface compared to the dielectric permittivity of the air. Arc segment bounded by an angle $\Theta$ is chosen to be equal to the half of wavelength of the creeping wave ($\lambda_{\text{eff}}/2$). The length of the tangential segment bounded by the same angle $\Theta$ is defined as the half wavelength of the wave in the free space ($\lambda_0/2$).

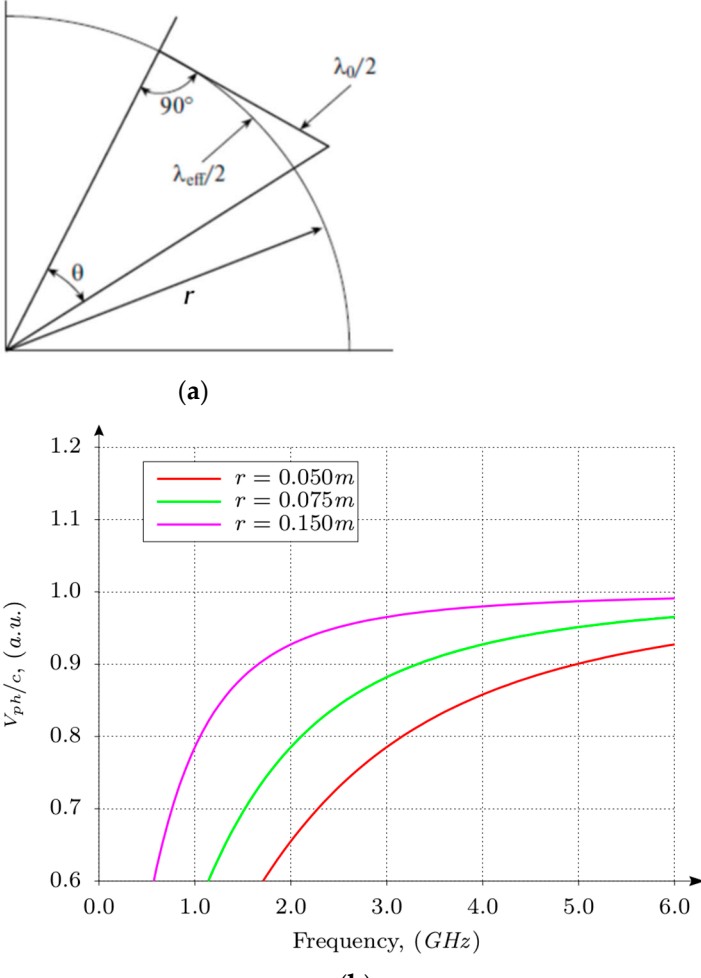

**(a)**

**(b)**

**Figure 17.** (**a**) Simplified geometry of a shoulder for creeping wave phase velocity modeling; (**b**) Analytical frequency dependence of the normalized phase velocity.

Actually, the following inequalities are fulfilled: $\lambda_{eff} < \lambda_0 << r$. In this case, the wavelength of the CW and the wavelength in the free space are defined as

$$\frac{\lambda_{eff}}{2} = \Theta \cdot r \tag{13}$$

$$\frac{\lambda_0}{2} = r \cdot \tan \Theta \tag{14}$$

It follows from Equations (13) and (14):

$$\lambda_{eff} = 2 \cdot r \cdot \tan^{-1}\left(\frac{\lambda_0}{2 \cdot r}\right) \tag{15}$$

The phase velocity at the frequency $f$ is described by the equation:

$$\frac{v_{eff}}{f} = 2 \cdot r \cdot \tan^{-1}\left(\frac{c}{2 \cdot f \cdot r}\right) \tag{16}$$

followed by the normalized effective phase velocity [43]:

$$\frac{v_{eff}}{c} = \frac{2 \cdot f \cdot r}{c} \cdot \tan^{-1}\left(\frac{c}{2 \cdot f \cdot r}\right) \tag{17}$$

The frequency dependence of the normalized phase velocity of the CW using Equation (17) is presented in Figure 18 together with full-wave simulation results.

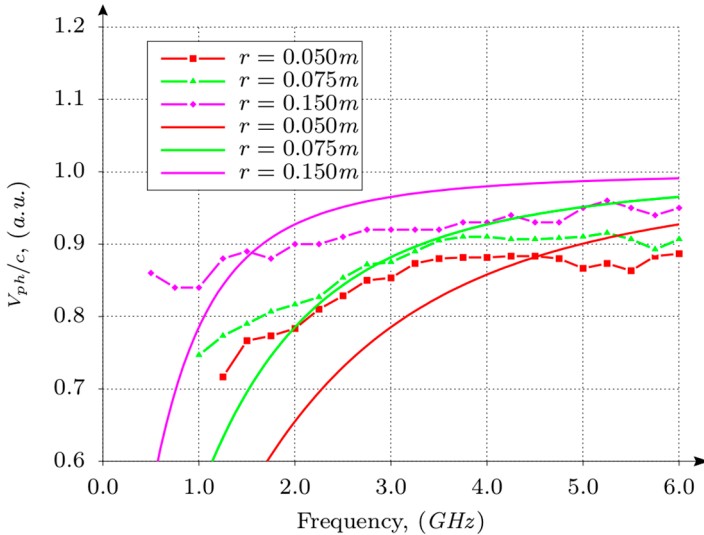

**Figure 18.** A comparison of the analytical model and full-wave simulations of creeping wave phase velocity.

The experimental study of the wave propagation over a human body was performed on the volunteer (Figure 19) to obtain the most accurate results that are close to realistic phenomena as much as possible. All results were obtained at the frequency $f$ = 2.55 GHz [18,43,44].

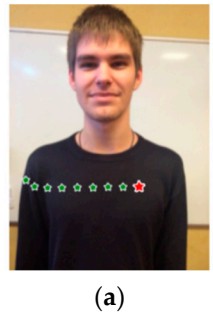

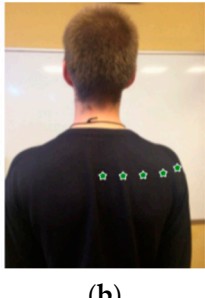

(a)                                                      (b)

**Figure 19.** (**a**)The antenna position over the human body. (**b**)The position of the stationary transmitting antenna is shown by the red star; the green stars show the receiving antenna positions.

Two identical disk antennas (Figure 20) of 70 mm in diameter were fabricated and tested to estimate the propagation loss. The antennas are well matched at the frequency $f$ = 2.55 GHz, which is confirmed by the measurement of the reflection coefficient. To measure the transmission coefficient between antennas, the antennas were positioned at a height of 5 mm over the body surface. The radiating antenna was placed at the left part of the chest (marked with the red star in Figure 19) while the receiving antenna was being moved along the chest, then around the right shoulder and along the back in the lateral plane. The positions of the receiving antenna are marked in Figure 19 with the green stars. The transmission coefficient (propagation loss) was measured for each position of the receiving antenna. Measured and simulated results are presented in Figure 21, where the SW paths (chest and back) propagation loss is presented by the blue and green curves, and the CW path is presented by the curve colored in magenta. The SW propagation loss on the back was taken from the modeling results shown in Figure 15b, whereas the SW propagation loss on the chest was obtained by the parallel shifting of the SW loss on the back since the dependence of the loss with distance is the same except the initial value at the first point on each of the paths. A good agreement is observed between the

experimental and computational results. A couple of experimental points belonging to the back of the body do not go well with the modeling results. It could happen because they were influenced by the wave leakage over the shoulder radiated from the transmitting antenna in the vertical direction.

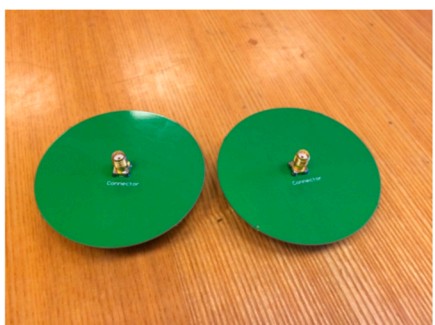

**Figure 20.** Two manufactured disc antennas used for on-body propagation loss measurements.

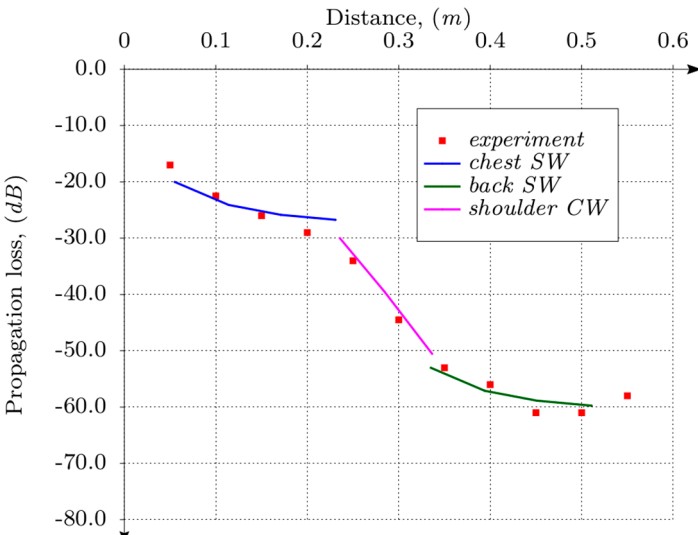

**Figure 21.** Measured insertion loss between the antennas along the path on a body. The distance between the antennas is shown on the horizontal axis, and the insertion propagation loss is shown on the vertical axis. Red dots represent the measured receiving antenna positions. The blue and green line is the predicted SW propagation loss from the modeling results presented in Figure 15a, and the magenta line is the predicted CW propagation loss from the modeling results presented in Figure 15b.

## 4. Conclusions

The results of theoretical and experimental studies of the propagation of EM waves inside of the human body and on its surface allow the formulation of the following conclusions:

The total internal reflection at the interface between the two dielectric layers and the dielectric loss in the biological tissue is the main contributions to the attenuation of the wave propagating in the biological medium.

The problem of refraction on the boundary between the biological medium and the free space is analytically solved, taking into account the dielectric loss. The proposed analytical consideration is in good agreement with the results of the electrodynamics simulation of the attenuation between the antenna inside the homogeneous tissue and the external antenna located outside the tissue.

A reliable estimation of parameters of different parts of the human body, namely, frequency-dependent dielectric permittivity and electric conductivity of tissues, provides a correct description of the EM propagation inside and along the body.

A model of a body considered as a perfect conductor can be used to simulate the propagation of EM waves in the microwave range over the body surface using both numerical and analytical methods; the accuracy of the performed calculations is high and suitable for practical applications.

Analytical and numerical modeling of the propagation loss of EM waves for different types of the surface (flat or curved) provides an accurate calculation of the loss-factor as a function of the distance traveled by the wave.

An investigation of the dispersion characteristics of the surface and creeping waves has been carried out; the identified dispersion of the creeping wave should be taken into account in practical applications.

An investigation of the EM wave propagation inside the multilayer biological object demonstrates a possibility to use these waves for microwave diagnostics of the state of different parts of the human body.

The results presented may be used for monitoring the characteristics of different parts of the human body considered as fundamentals of diagnostics and other medical applications. The main goal of further research is the improvement of the experimental technique in order to obtain information in vivo about biological tissue characteristics using the experience of studying the propagation of EM waves in a multilayer structure of complex geometry.

**Author Contributions:** I.V.—writing: original draft preparation, review and editing the manuscript; O.V. contributed in analytical model development and analysis of EM wave propagation on the human body surface; V.P. contributed in analysis of EM wave propagation in multilayer object and investigation of on-body wave propagation; I.M. investigated the EM wave propagation through the biological medium–air interface; P.T. investigated the EM wave propagation in the regular biological medium and reflection and refraction of the waves on the body–air interface; V.K. took part in all experimental investigations and the result analysis. All authors have read and agreed to the published version of the manuscript.

**Funding:** This research received no external funding.

**Conflicts of Interest:** The authors declare no conflict of interest.

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
