# Peer review of "Wireless Monitoring of Biological Objects at Microwaves"

_electronics, doi:10.3390/electronics10111288_

Round 1

Reviewer 1 Report

The paper is a thoroug overview of EM propagation inside and along the surface of the human body. Experimental results are also provided for a miniature implanted antenna, EM propagation across human bone and finally CW and SW surface propagation.

I have some comments as listed below:

  • The quality of almost all figures are poor, they should be improved. Sometimes the text on the figures are not readable.
  • It is not denoted if a figure is not an own one. I suppose that there are several third party figures (Fig.2., Fig.3., Fig.10, Fig.11?, 
  • In the text, especially after the equations the parameters with subsciptons are not using subscripted fonts.
  • In 2.1 please specify what you mean 'low frequency'.
  • For equations provided, please apply a reference if the equation is not own.
  • In line 209 there is a typo.
  • For line 214 the selected frquency is in the ISM band, it is recommended to mention this.
  • Figure 6a is showing the implanted antenna. Please specify the dimensions.
  • In line 270 there is a typo.
  • In Figure 8b the two line colors and style seems to be identical.
  • It is not clear if the results shown in Fig. 12 is are own measurements or not. The dimensions of the printed antenna in Fig.12a is not provided.
  • The goal of the experimental study of propagation EM waves over human body is not clear. What is the application area, or how the results can be used? Please clarify this.
  • In Fig.20 the antenna dimensions are not given.
  • In Fig.21 it is unclear what is the distance, between which objevts and how was measured? Also, I suppose that the blue and purple lines in the figure were only drawn by hand. This should be clarified.

Author Response

Authors would like to thank the reviewer for the valuable comments; for sure it helped in improving the research quality.

  • The quality of almost all figures are poor, they should be improved. Sometimes the text on the figures are not readable.

The quality of all figures is improved. The text is now readable.

  • It is not denoted if a figure is not an own one. I suppose that there are several third party figures (Fig.2., Fig.3., Fig.10, Fig.11),

References to the ownership of the Fig.2., Fig.3., Fig.10, Fig.11 by the authors of the paper are given in the text of the paper

  • In the text, especially after the equations the parameters with subscriptions are not using subscripted fonts.

The parameters with subscriptions are corrected in the text.

  • In 2.1 please specify what you mean 'low frequency'.

The low frequencies are specified as f < 1 MHz,

  • For equations provided, please apply a reference if the equation is not own.

The reference is added to the eq. [4]-[10], [11], [17]

  • In line 209 there is a typo.

Corrected

  • For line 214 the selected frequency is in the ISM band, it is recommended to mention this.

Mentioned:  f = 915 MHz (ISM band)

  • Figure 5a is showing the implanted antenna. Please specify the dimensions.

The dimensions are given in the text: The dipole with area of 20x10 cm

  • In line 270 there is a typo.

Corrected

  • In Figure 8b the two line colors and style seems to be identical.

Corrected

  • It is not clear if the results shown in Fig. 12 are own measurements or not.

References to the ownership of the fig. 12 by the authors of the paper are added to the figure caption: Figure 12. Electric field distribution in the finger excited by the printed antenna: printed antenna placed on top of the finger (a), electric field distribution inside the modeled tissues at 750 MHz (b), and at 2225 MHz (c); return loss of the antenna: red curves – diseased bone, green curves – healthy bone, solid lines – modeling, dashed lines – experiment (d) [42]-[44].

The dimensions of the printed antenna in Fig.12a is not provided.

The dimensions of antenna are given in the text: antenna with dimensions of 5.5x0.35 cm

  • The goal of the experimental study of propagation EM waves over human body is not clear. What is the application area, or how the results can be used? Please clarify this.

The explanation is added to the very first paragraph in section 3.

“Study of surface wave propagation mechanism is important in situations when the wireless data transfer between sensors placed on the human body surface is in need. In some situations it is not feasible or impractical to connect by wire sensors like bio-potential or acoustic readers or execution devices like injection pumps, with a computer running some software intended for interaction with the body. A good understanding of the wireless data channel properties is the key aspect of the wireless connection successful operation. Wireless links running on the surface of the human body have specific properties depending on the body dimensions, shape and surface impedance not to mention the influence of other factors like individual variations in the surface impedance and presence or absence of clothes, etc.”

  • In Fig.20 the antenna dimensions are not given.

The antenna diameter was given in the text:

“… disk antennas (Figure 20) of 70 mm in diameter…”

  • In Fig.21 it is unclear what is the distance, between which objects and how was measured? Also, I suppose that the blue and purple lines in the figure were only drawn by hand. This should be clarified.

The clarifications are added to the main text and to the caption in Figure 21.

Figure 21. Measured insertion loss between the antennas along the path on body. The distance between the antennas is shown on horizontal axis and the insertion propagation loss is shown on vertical axis. Red dots repersent the measured receiving antenna positions. Blue and green line is the predicted SW propagation loss from the modeling results presented in Figure 15 (a) and the magenta line is the predicted CW propagation loss from the modeling results presented in Figure 15 (b).

Reviewer 3 Report

Electromagnetic (EM) wave propagation inside and on the surface of the human body is the subject of current research in both biomedical applications and microwave technology. This area of ​​research is the basis for wireless monitoring of parameters and characteristics of biological objects. Solving communication problems is essential for achieving the set goals in the field of wireless monitoring on the human body. Monitoring of biological objects is based on consideration of a defined range of problems. The results of a study of theoretical and experimental approaches to the propagation of EM waves inside and on the surface of the human body have provided a reliable estimate of the parameters of different parts of the human body, frequency-dependent dielectric and conductivity parameters and thus provides a correct description of EM propagation inside and along the body. Models of the human body have been designed and verified, and analytical and numerical models of EM wave propagation losses for various types of simple surfaces (flat or curved) have been modeled. The work is scientifically conceived, brings new knowledge to specific questions and is processed with the use of experiment.

The same work appears twice in the text under references 13 and 57. I recommend editing. 

Vendik, I. B., Vendik, O. G., Kozlov, D. S., Munina, I. V., Pleskachev, V. V., Rusakov, A. S., & Tural’chuk, P. A. (2016). Wireless monitoring of the biological object state at microwave frequencies: A review. Technical Physics, 61(1), 1-22. doi:10.1134/S1063784216010242 

I recommend for publication. 

Author Response

Authors would like to thank the reviewer for the  positive estimation of the paper.

The only comment was formulated by the Reviewer:

"The same work appears twice in the text under references 13 and 57. I recommend editing. 

Vendik, I. B., Vendik, O. G., Kozlov, D. S., Munina, I. V., Pleskachev, V. V., Rusakov, A. S., & Tural’chuk, P. A. (2016). Wireless monitoring of the biological object state at microwave frequencies: A review. Technical Physics, 61(1), 1-22. doi:10.1134/S1063784216010242 "

Response: In the revised version of the paper, there is no reference 57.

Reviewer 4 Report

The work presents new contributions, treating both by simulation (commercial numerical software) and by measurements, the topics of  microwave imaging. Propagation peculiarities when emitting antennas are situated inside the body - near to the surface, or on the body, are carefully analyzed and their limitations are described and surpassed. The final object is improving microwave imaging and providing individualized methods to investigate health state of tissues with minimal hazards procedures based on dielectric properties and creeping wave behavior.

All the observations and suggestions to the authors are found on the annotated manuscript which is attached here. Not major deficiencies were observed, so that the paper can be rapidly improved.

Author Response

Authors would like to thank the reviewer for the  positive estimation of the paper.

There are no comments for the text editing.